# Zika Virus Induces Degradation of the Numb Protein Required through Embryonic Neurogenesis

**DOI:** 10.3390/v15061258

**Published:** 2023-05-27

**Authors:** Jia He, Liping Yang, Peixi Chang, Shixing Yang, Yu Wang, Shaoli Lin, Qiyi Tang, Yanjin Zhang

**Affiliations:** 1Molecular Virology Laboratory, Department of Veterinary Medicine, University of Maryland, College Park, MD 20742, USA; hejia86@hotmail.com (J.H.); pingzongyunlan@gmail.com (L.Y.); changpxi@umd.edu (P.C.); johnsonyang1979@163.com (S.Y.); wangyu1974-1@163.com (Y.W.); linshaoli19903@gmail.com (S.L.); 2Department of Microbiology, Howard University College of Medicine, Washington, DC 20059, USA; qiyi.tang@howard.edu

**Keywords:** Zika virus, ZIKV, the Numb protein, the capsid protein

## Abstract

Zika virus (ZIKV) is a mosquito-borne flavivirus and causes an infection associated with congenital Zika syndrome and Guillain–Barre syndrome. The mechanism of ZIKV-mediated neuropathogenesis is not well understood. In this study, we discovered that ZIKV induces degradation of the Numb protein, which plays a crucial role in neurogenesis by allowing asymmetric cell division during embryonic development. Our data show that ZIKV reduced the Numb protein level in a time- and dose-dependent manner. However, ZIKV infection appears to have minimal effect on the *Numb* transcript. Treatment of ZIKV-infected cells with a proteasome inhibitor restores the Numb protein level, which suggests the involvement of the ubiquitin–proteasome pathway. In addition, ZIKV infection shortens the half-life of the Numb protein. Among the ZIKV proteins, the capsid protein significantly reduces the Numb protein level. Immunoprecipitation of the Numb protein co-precipitates the capsid protein, indicating the interaction between these two proteins. These results provide insights into the ZIKV–cell interaction that might contribute to its impact on neurogenesis.

## 1. Introduction

Zika virus (ZIKV) is a mosquito-borne virus in the genus *Flavivirus*, the *Flaviviridae*. ZIKV was first isolated in 1947 from a rhesus monkey in Africa [1,2]. ZIKV is enveloped and has a positive-sense, single-stranded RNA genome of 10.7 kb in length. It has the common feature of Flavivirus that the genome encodes a single polyprotein, which is cleaved into individual products, including structural proteins (capsid (C), precursor membrane (prM), and envelope (E)) and non-structural proteins (NS1, NS2A, NS2B, NS3, NS4A, NS4B, and NS5) [3]. ZIKV strains are phylogenetically grouped into African lineage and Asian lineage. ZIKV infects neural precursor cells derived from pluripotent stem cells and causes apoptotic cell death and cell-cycle dysregulation [3,4,5]. An Asian lineage ZIKV strain infects embryonic mouse brains and causes microcephaly [6,7]. There was little attention paid to ZIKV infection until the first documented human outbreak in the Pacific islands in the late 2000s and the later epidemic in South America in 2015–2016 [8,9,10,11]. The ZIKV strains in the recent outbreaks belong to the Asian lineage and are associated with congenital Zika syndrome (microcephaly, cerebral calcifications, and macular scarring) and Guillain–Barre syndrome (GBS) [11,12,13,14,15,16]. There is no approved effective treatment or vaccine to control ZIKV infection [17].

Earlier research indicates that ZIKV interacts with critical cellular proteins and leads to neuropathogenesis through multiple mechanisms, including apoptosis, premature differentiation, and inflammatory response [18,19]. However, ZIKV–cell interactions in the aspect of neuropathogenesis are not well understood. Because of ZIKV’s negative impact on neonatal neurological development, we were interested in ZIKV’s interaction with the critical cellular proteins involved in neurogenesis, among which the Numb protein is well recognized in its unique role in this process. The *Numb* gene was initially discovered in Drosophila for its determination of cell fate in sensory neuron formation [20]. The function of the Numb protein in cell fate determination is best studied in *Drosophila* sp. [21]. During neurogenesis, the Numb protein localizes to one side of the progenitor cell and selectively segregates into one daughter cell. The asymmetric division leads to one daughter cell generally differentiating into a neuron cell and the other becoming a progenitor for further proliferation. Its homolog in humans is encoded by the *Numb* gene, which is well-conserved from invertebrates to mammals. Knockout of the *Numb* gene is lethal to the mouse at the early embryo stage [22]. Our study aimed to determine the ZIKV effect on *Numb* expression. We noted the Numb protein was decreased significantly in ZIKV-infected cells in a time and dose-dependent manner. Further study was conducted to examine the mechanism of the ZIKV-mediated reduction in the Numb protein. Our results provide insights into the ZIKV–cell interactions that may contribute to a better understanding of ZIKV-induced neuropathogenesis.

## 2. Materials and Methods

### 2.1. Viruses, Cells, and Chemicals

ZIKV PRVABC59 strain (ATCC VR-1843; GenBank Accession Number KX377337) and ZIKV MR766 strain (ATCC VR-1838; GenBank Accession Number NC012532.1) were used to infect Vero cells. Virus yields were titrated in Vero cells to obtain the median tissue culture infectious dose (TCID_50_) [23].

HEK293 (ATCC CRL-1573), HeLa (ATCC CCL-2), Vero (ATCC CCL-81), and SK-N-SH (ATCC HTB-11) cells were maintained in Dulbecco Modified Eagle Medium (DMEM) supplemented with 10% fetal bovine serum at 37 °C and 5% CO_2_. 

Proteasome inhibitor MG132 (Enzo Life Sciences, Farmingdale, NY, USA) was applied to cultured cells at a working concentration of 10 µM for 6 h before the cells were harvested. Lysosome inhibitor NH_4_Cl was applied to cultured cells at a working concentration of 10 mM for 6 h prior to cell harvesting. A protein translation inhibitor, Cycloheximide (Fisher Scientific, Waltham, MA, USA, AC357420010), was used at a final concentration of 50 μg/mL to ZIKV-infected and mock-infected cells to determine the half-life of Numb. 

CellTiter-Glo^®^ Luminescent Cell Viability Assay Kit (Promega, Madison, WI, USA, G7570) was used to determine cell viability by following the manufacturer’s instructions. 

### 2.2. Plasmids

The cloning of ZIKV genes for the expression of individual proteins was previously described [24]. The Numb plasmid was constructed by cloning the Numb-coding sequence into the pCAGEN-Myc vector after PCR amplification with primers Numb-F1 and Numb-R1 (Table 1) on cDNA prepared from total RNA of Vero cells. 

For shRNA against *Numb*, the oligos for shRNA were cloned into the pSIREN-RetroQ-ZsGreen vector (Clontech, Mountain View, CA, USA, 632455) following the manufacturer’s instruction. GP2-293 cells (Clontech, 631458) were used for retrovirus packaging. pSIREN-RetroQ-ZsGreen-shNumb and VSV-G vector (a gift from Akitsu Hotta (Addgene plasmid # 138479; http://n2t.net/addgene:138479; accessed on 15 May 2023; RRID:Addgene_138479) [25]) plasmids were co-transfected into GP2-293 cells. The preparation of control shRNA (C-shRNA) plasmid was described previously [24]. The GP2-293 culture supernatant containing the recombinant retrovirus was used to transduce Vero cells. 

### 2.3. RNA Isolation and Real-Time PCR

TRIzol™ reagent (Thermo Fisher Scientific, 15596026) was used for extracting total RNA from Vero cells following the manufacturer’s instructions. Reverse transcription and real-time PCR (RT-qPCR) with SYBR Green detection (Thermo Fisher Scientific, 4334973) were performed as described previously [26,27,28]. The transcripts of *RPL32* (ribosomal protein L32), a housekeeping gene, were also determined. The relative transcript levels were shown as folds compared with the control cells after *RPL32* normalization. The real-time PCR primers used for *Numb* were Numb-RR-F1 and Numb-RR-R1, while those for ZIKV were ZIKV-RR-F and ZIKV-RR-R (Table 1). For the absolute quantitation of ZIKV RNA copies in the infected cells, a standard curve using the NS5 plasmid [24] as a template was included. The RNA copies were calculated based on the total amount of RNA isolated from one well of a 24-well plate and shown as the total ZIKV RNA level. All experiments were repeated at least three times, each conducted in triplicate. 

### 2.4. Western Blot (WB) Analysis

The Laemmli sample buffer was used in cell lysis to prepare whole cell lysates for SDS-PAGE and WB, as described previously [24,28,29]. The primary antibodies against Numb (Santa Cruz Biotechnology, Santa Cruz, CA, USA, sc-136554), Myc tag (Santa Cruz Biotechnology, sc-40), GFP (Biolegend, San Diego, CA, USA, 75818-584), ubiquitin (Santa Cruz Biotechnology, sc-8017), GAPDH (Santa Cruz Biotechnology, sc-365062), TUBB1/β-tubulin (Sigma, Livonia, MI, USA, T7816), ZIKV C (GeneTex, Irvine, CA, USA, GTX133317), ZIKV E (B.E.I. Resources, Newport News, VA, USA, NR-50413), ZIKV NS4B (GeneTex, GTX133311), and ZIKV NS5 (GeneTex, GTX133329) were used in this study. Goat anti-mouse or anti-rabbit IgG conjugated with horseradish peroxidase (Bio-Rad, Hercules, CA, USA, 170–5046, and 170–5047) were used as secondary antibodies in this study. For revealing the specific reactions, a chemiluminescence substrate was used, and the signal was recorded digitally using a Bio-Rad ChemiDoc XRS imaging system with the QuantityOne Program, version 4.6 (Bio-Rad Laboratories, CA). Densitometry analysis of the WB images was performed with the QuantityOne Program, version 4.6 (Bio-Rad). All WB images were acquired in the linear range of digital intensity without saturated pixels. 

### 2.5. Immunoprecipitation (IP)

Immunoprecipitation was performed as described previously [24]. Cell lysates were clarified and incubated with specific antibodies indicated in the results or figure legends, followed by incubation with protein A/G-magnetic beads (Bimake.com, B23202). The IP complexes were subjected to WB for the detection of target proteins indicated in the results or figure legends.

### 2.6. Statistical Analysis

Differences in gene expression between the treatment group and control were assessed using Student’s *t*-test in GraphPad Prism 5. A two-tailed *p*-value of 0.05 was considered significant.

## 3. Results

### 3.1. ZIKV Infection Reduces the Numb Protein Level

Vero cells were infected with the ZIKV PRVABC59 strain (Asian lineage) at a multiplicity of infection (MOI) of 10 [24,28]. Western blotting was performed to detect the Numb protein level. The results showed that the Numb protein level decreased by 90% in the ZIKV-infected cells at 40 h post-infection (hpi) compared with the mock-infected cells (Figure 1A). To investigate if the African lineage ZIKV has a similar effect on the Numb protein level, we infected Vero cells with ZIKV MR766 strain. MR766 strain infection reduced the Numb protein level by 70% at 40 hpi (Figure 1B). 

The results above showed the Numb protein reduction in ZIKV-infected Vero cells. To exclude the possibility that Numb reduction is cell-line specific, we determined Numb protein levels in other cell lines, including HeLa cells and neuroblastoma SK-N-SH cells. The latter cell line is more physiologically relevant to ZIKV brain infection and its associated neurological disorders. The results showed that the PR strain also reduced Numb in both HeLa cells and SK-N-SH cells (Figure 1C,D). These results demonstrate that ZIKV infection reduces the Numb protein level. 

### 3.2. ZIKV Reduces the Numb Protein Level in a Temporal and Dose-Dependent Manner

As ZIKV infection led to Numb protein reduction, we reasoned that the Numb protein reduction would be in a temporal manner. To test this, we infected Vero cells with PR strain at an MOI of 1 and harvested the cells for WB. The results showed that ZIKV reduced the Numb protein level at 20 and 40 hpi by 30% and 80%, respectively, in comparison with mock-infected control (Figure 2A). In addition, Vero cells were infected with ZIKV PR strain at MOIs of 0.1, 1, 5, and 10 and harvested for WB 24 hpi. The results showed that the Numb protein decreased further along with the incremental MOI (Figure 2B). ZIKV NS4B protein was detected (Figure 2B), and the ZIKV RNA level was determined with reverse transcription and real-time PCR (RT-qPCR) (Figure 2C), which confirmed the increase in ZIKV replication along with the incremental inoculum. 

### 3.3. ZIKV Reduces the Numb Protein via the Ubiquitin–Proteasome Pathway

The reason for a protein reduction in cells could be lower transcription, translation, or protein stability. For the reduction in the Numb protein level, we wondered if ZIKV had any effect on the Numb transcript. Total RNA was isolated from ZIKV-infected cells and *Numb* mRNA was determined by RT-qPCR. The results showed that the *Numb* mRNA level in the ZIKV-infected cells was similar to that in the mock-infected cells (Figure 3A). ZIKV RNA level was determined to confirm the infection (Figure 3B). These results indicate that the Numb protein reduction caused by ZIKV is not due to the inhibition of transcription. There are two primary organelles involved in protein degradation in cells: the proteasomes and the lysosomes [30,31].

Next, we intended to determine which of the two organelles was responsible for the Numb protein degradation using two chemicals: MG132 and ammonium chloride (NH_4_Cl). MG132 is a proteasome inhibitor blocking the degradation of proteins in proteasomes in the cells. Lysosomotropic compound NH_4_Cl raises the lysosomal pH, leading to blockage of proteolysis in the lysosomes [32]. At 6 h before harvesting, the cells were treated with MG132 or NH_4_Cl. The results showed that treatment with MG132 restored the Numb protein level in the ZIKV-infected Vero cells, while NH_4_Cl failed to do so (Figure 3C). The result suggests that the ZIKV-mediated Numb reduction occurs via the ubiquitin–proteasome pathway instead of the lysosomal proteolysis. We then determined the Numb protein half-life in the ZIKV-infected cells, as degradation led to the Numb protein reduction. Cycloheximide, an inhibitor of protein translation, was added to both mock-infected and ZIKV-infected Vero cells. MG132 was used to treat the infected cells at 24 hpi for 6 h before the addition of cycloheximide to shore up the Numb protein level for this test. The cells were harvested at 0, 6, 12, 24, and 36 h after the cycloheximide treatment for WB. The result shows that the Numb protein level was reduced much faster in the infected cells than in mock-infected cells and the densitometry analysis indicates that ZIKV infection shortened the Numb half-life from 36 h to 12 h (Figure 3D). This result suggests that ZIKV reduces Numb protein stability by accelerating its degradation. 

The results above showed the Numb protein reduction caused by ZIKV infection, possibly via the ubiquitin–proteasome pathway. We reasoned that the ZIKV infection would induce the Numb protein polyubiquitination, which is needed for target protein degradation via this pathway. To confirm this speculation, we conducted immunoprecipitation of the Numb protein, followed by WB with antibodies against ubiquitin and Numb. The results showed that the Numb ubiquitination level in the ZIKV-infected cells was significantly elevated by 2.75-fold in comparison with the mock-infected cells (Figure 3E). A similar total ubiquitination level was observed between the infected and mock-infected cells. Together, these results indicate that ZIKV infection induced Numb polyubiquitination and degradation via the proteasomes. 

### 3.4. ZIKV Capsid Protein Induces the Numb Reduction

ZIKV genome encodes a single polyprotein, which is cleaved into structural proteins (C, prM, and E) and non-structural proteins (NS1, NS2A, NS2B, NS3, NS4A, NS4B, and NS5) [33]. To determine which of these ZIKV proteins induces the Numb protein reduction, we transfected HEK293 cells with plasmids encoding the individual proteins of ZIKV [24] and a plasmid encoding the Numb protein. Compared with the empty vector control, the ZIKV C protein induced a significant reduction in the Numb protein level by 99.5%, while the other proteins had much less or minimal effects (Figure 4A). Therefore, we selected the C protein for further study. 

As ZIKV C induces the reduction in Numb, we speculated that the two proteins might have an interaction. To test the speculation, we infected Vero cells with ZIKV and conducted IP with the antibody against the Numb protein. The WB results showed that the Numb IP co-precipitated the ZIKV C protein (Figure 4B). To exclude the possibility of other viral proteins involved in the interaction, we transfected HEK293 cells with the plasmids encoding the C and the Numb protein and harvested the cells 24 h later for co-IP and WB. The results showed that the ZIKV C protein was present in the Numb co-IP precipitates (Figure 4C). These results indicate that the ZIKV C protein induces the Numb reduction and interacts with the Numb protein. 

### 3.5. The Numb Knockdown Has a Minimal Effect on ZIKV Replication

The results above displayed ZIKV-induced Numb reduction via the ubiquitin–proteasome pathway. We wondered if the Numb protein had any effect on ZIKV replication. To explore the role of the Numb protein in ZIKV proliferation, we conducted RNAi-mediated knockdown of *Numb* expression in Vero cells using recombinant retrovirus expressing shRNA against *Numb* mRNA. A control shRNA (C-shRNA) of an irrelevant sequence was included in the study. The Numb protein level in the cells with shNumb was significantly reduced in comparison with the cells with C-shRNA (Figure 5A). ZIKV infection further reduced the Numb protein level. To determine the ZIKV yield in the cells with C-shRNA or shNumb, we collected cell culture supernatant samples at 24, 48, 72, 96, and 120 hpi for virus titration. The results showed that the ZIKV viral yields in the cells with the Numb knockdown were similar those in the cells with C-shRNA (Figure 5B). The Numb knockdown had minimal effect on the cell growth (Figure 5C). These results showed that the Numb knockdown had minimal effect on ZIKV replication in Vero cells.

## 4. Discussion

Our data demonstrate that ZIKV infection reduces the Numb protein level via the ubiquitin–proteasome pathway and that the ZIKV capsid protein induces Numb degradation. Intriguingly, the *Numb* knockdown has minimal effect on ZIKV replication, which suggests that the ZIKV-mediated Numb reduction is probably related to ZIKV pathogenesis, especially considering the Numb’s role in embryonic neurogenesis. 

To determine the ZIKV effect on Numb expression, we used three cell lines and two strains of the virus to confirm the virus-mediated reduction. The Asian-lineage PR strain appears to induce a greater and quicker Numb reduction than the African-lineage MR766 strain, which is potentially consistent with the clinical manifestation that the Asian-lineage strains are associated with neurological manifestations. There are four different amino acids in the capsid protein between the PR and MR766 strains. However, it is not known whether the different amino acid residues account for the difference in the Numb reduction. The cell lines tested include SK-N-SH, derived from neuroblastoma cells, which is more physiologically relevant to ZIKV infection of neural cells. Our results show that ZIKV infection reduces the Numb protein in SK-N-SH cells at 72 hpi, which may be because of the slower cell growth than Vero and HeLa cells and, subsequently, lower ZIKV replication in this cell line. The ZIKV-mediated reduction in Numb is shown to be temporal and dose-dependent, which indicates the virus-specific effect. 

ZIKV induces Numb degradation via the ubiquitin–proteasome pathway, shown by the restoration by MG132 but not NH4Cl treatment. This test eliminates the potential involvement of the lysosomes as MG132 also targets certain hydrolases in the lysosomes, whereas NH4Cl is lysosomotropic [32]. We further showed the half-life reduction in the Numb protein and its elevated polyubiquitination by ZIKV infection, which provides further evidence that ZIKV reduces Numb via the ubiquitin–proteasome pathway. Our data indicate the Numb half-life is 36 h in Vero cells, which is much longer than 10 h in C2C12 myoblasts in an earlier report [34]. This discrepancy is possibly caused by the different cell types, as C2C12 myoblasts are stem cells that differentiate into muscular cells, while Vero cells are derived from the kidney epithelial cells of an African green monkey. 

Our screening of ZIKV proteins showed that the C protein is primarily responsible for the Numb reduction. We noted that NS2B reduced the Numb protein level by 50%, suggesting it may play a minor role in the Numb reduction. Intriguingly, NS2B3, NS4A, NS4B, and NS5 could induce an increase in the Numb level in transient expression. This observation does not corroborate with the results of the whole virus infection, and we did not pursue it further. Multiple bands were observed for several ZIKV proteins, including prM, NS3, and NS4A, in the transiently transfected cells, which suggests that post-translational cleavage might have happened to them. In determining the mechanism of the C-mediated Numb reduction mechanism, we noted that Numb IP could co-precipitate the C protein in both ZIKV-infected and transiently co-transfected cells. This result suggests that the Numb protein interacts with the C protein in the absence of other viral proteins. The ZIKV C protein is known to bind the viral genomic RNA and is involved in virion assembly [35]. Besides virion assembly, the C protein is also known to interact with cellular proteins to play roles in assisting ZIKV replication, modulating cellular metabolism, and antiviral response. The C protein induces the loss of peroxisomes, which have an important role in innate immunity [36]. It targets the nonsense-mediated mRNA decay (NMD) pathway, a cellular mRNA surveillance mechanism, via interacting with up-frameshift protein 1 (UPF1), a central NMD regulator, and targeting it for degradation [37]. The disruption of the NMD pathway may contribute to neuropathology. In a mosquito cell line stably expressing C, 157 interactors were identified, and 8 have proviral activity during ZIKV infection in the cells, including transitional endoplasmic reticulum protein TER94 [38]. ZIKV inhibits the formation of stress granules (SGs), and C interacts with SG components G3BP1 and Caprin-1 [39]. The ZIKV C, but not other flaviviruses, antagonizes endoribonuclease Dicer and, consequently, inhibits miRNA biogenesis in neural stem cells, which leads to the disruption of corticogenesis [40]. Mutant C (H41R) loses interaction with Dicer, and ZIKV-H41R does not inhibit neurogenesis. These data demonstrate that the C protein plays an important role in ZIKV replication and invasion/pathogenesis. Our finding of C-mediated Numb reduction further contributes to the literature on the functions of this essential protein. 

The effect of the Numb protein on virus infection is rarely studied and appears to be virus-dependent. The Numb protein is needed for hepatitis C virus entry as RNAi-mediated knockdown of the *Numb* inhibits the virus entry [41]. However, Numb inhibition activates Notch signaling, which assists the transcription of covalently closed circular DNA (cccDNA) of the hepatitis B virus [42]. Our results of RNAi-mediated knockdown of *Numb* demonstrated that the Numb protein has no effect on ZIKV replication or cell growth of Vero cells. Taken together, our data suggest that the ZIKV-mediated Numb reduction might be relevant to its neuropathogenesis instead of viral replication. This is possible because Vero cells are differentiated and Numb appears not to contribute to the machinery needed for ZIKV replication. However, we cannot exclude the possibility that Numb might be needed for ZIKV replication in neural stem cells, because Numb reduction affects the fate of neural stem cell differentiation. Thus, if the total number of differentiated cells is reduced as a result of Numb downregulation, ZIKV proliferation might consequently be affected.

Numb functions in cell fate determination by antagonizing several developmental pathways, including Notch, Hedgehog, and WNT signaling, which are essential for regulation in cell proliferation, differentiation, cell fate determination, and self-renewal of stem cells and progenitor cells during embryonic development and in adult organs [43,44,45]. Numb inhibits Notch signaling via interaction with the Notch E3 ligase Itch and the Notch intracellular domain (NICD), leading to the degradation of NICD [21,43,46]. Numb suppresses the Hedgehog signaling pathway, a master regulator of embryonic development [47], by targeting the Hedgehog Gli1 transcription factor for Itch-dependent degradation [48]. Numb also inhibits the WNT pathway, another regulator of embryonic development, by promoting the degradation of β-catenin [49] and enhances p53 tumor suppressor activity by interacting with its E3 ligase MDM2 to block p53 degradation [50,51]. For these functions, Numb has been characterized as a tumor suppressor [21,52,53]. Indeed, a lower level of Numb correlates with a worse prognosis for several types of cancers. 

Numb is finely regulated at the transcriptional, translational, and post-translational levels. There are six functionally distinct isoforms of Numb ranging from 54 to 72 kDa in mammals owing to alternative splicing of *Numb* mRNA [54]. The Numb translation is suppressed by Musashi-1, an RNA-binding protein abundant in neural progenitor cells [55]. MicroRNA miR-146a inhibits Numb expression to regulate the proliferation and differentiation of muscle stem cells [56]. The Numb protein level is controlled by ubiquitin-dependent proteolytic degradation, which is mediated by E3 ligase LNX [57], MDM2 [51], and Siah-1 [58]. We assume that C potentially interacts with one of these E3 ligases and enhances its activity in Numb polyubiquitination. Indeed, a bioinformatic analysis of the ZIKV–host protein interaction network implies that the C protein interacts with MDM2 [59]. The mechanistic details of the ZIKV-C-mediated Numb reduction will be examined in future studies. 

In conclusion, our results demonstrate that ZIKV induces Numb reduction via the ubiquitin–proteasome pathway and that the C protein is responsible for the downregulation. The ZIKV-mediated reduction in the Numb protein level is expected to interfere with its functions, which is potentially relevant to ZIKV neuropathogenesis. Future studies are needed to address the contribution of Numb reduction to the ZIKV pathogenesis during fetal infection. Although the lack of confirmation in human neural cells is a limitation of this study, the data provide insight into ZIKV–cell interactions and may contribute to our better understanding of the molecular mechanism of ZIKV-induced neuropathogenesis. 

## Figures and Tables

**Figure 1 viruses-15-01258-f001:**
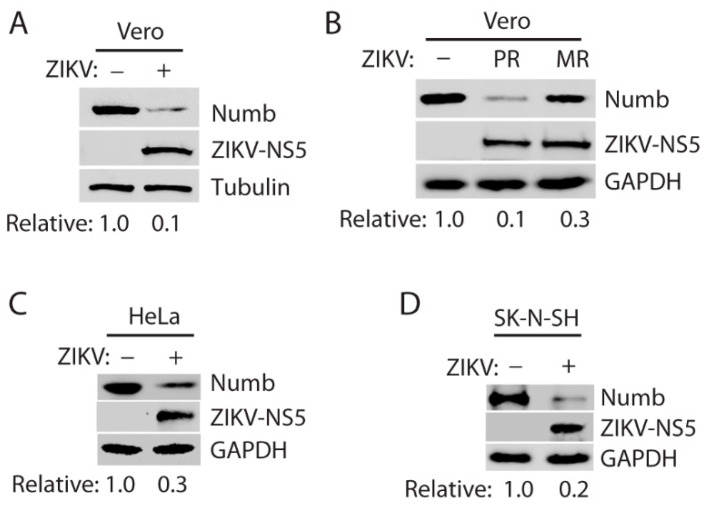
ZIKV infection reduces the Numb protein level. (**A**) Infection of Vero cells with ZIKV PRVABC59 (PR) strain leads to a lower Numb protein level. The cells were inoculated with an MOI of 10 of the ZIKV PR strain and harvested for Western blotting 40 h post-infection (hpi). Relative levels of Numb are shown below the images after normalization with tubulin. (**B**) Infection with ZIKV MR766 strain also induces a reduction in the Numb protein level. Vero cells were infected at an MOI of 10 and harvested at 40 hpi for WB. Relative levels of the Numb protein are shown below the images after normalization with GAPDH. (**C**) Infection of HeLa cells with ZIKV PR strain reduces the Numb protein. The cells were inoculated at an MOI of 10 and harvested for WB 36 hpi. (**D**) Numb protein level decreased in SK-N-SH cells with ZIKV infection. The cells were inoculated at an MOI of 10 and harvested for WB 72 hpi. Because of the variable progress in ZIKV proliferation in the different cell lines, the cells were harvested at different time points.

**Figure 2 viruses-15-01258-f002:**
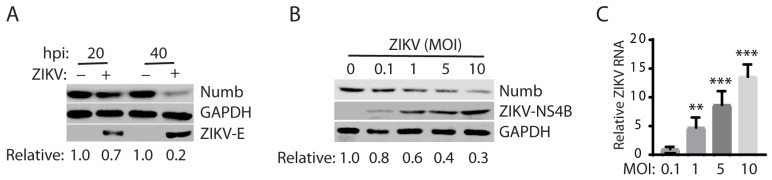
ZIKV reduces the Numb protein in a temporal and dose-dependent manner. (**A**) Temporal reduction in Numb by ZIKV infection. The cells were inoculated with ZIKV PR strain at an MOI of 1 and harvested for WB 20 and 40 hpi. Relative levels of the Numb protein are shown below the images. (**B**) Dose-dependent reduction in the Numb protein level by ZIKV PR strain. The cells were harvested for WB 24 hpi. Relative levels of the Numb protein are shown below the images. (**C**) ZIKV RNA levels in the infected cells detected by RT-qPCR. Error bars represent the standard errors of the means of three repeated experiments. Asterisks denote significant differences in RNA level from an MOI of 0.1 (** *p* < 0.01; *** *p* < 0.001).

**Figure 3 viruses-15-01258-f003:**
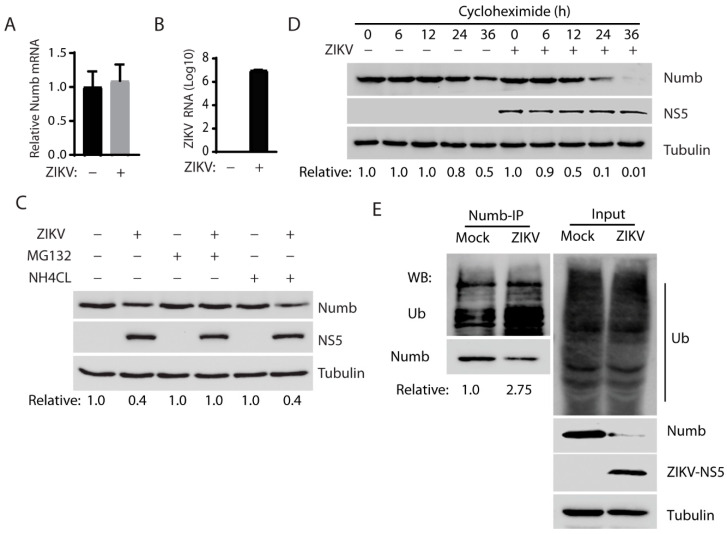
Numb was decreased by ZIKV infection through the ubiquitin–proteasome pathway. (**A**) ZIKV infection has a minimum effect on *Numb* mRNA level detected by RT-qPCR. The relative *Numb* mRNA level is shown compared with the mock-infected Vero cells. The cells were infected with the ZIKV PR strain at an MOI of 1 and harvested at 30 hpi. (**B**) ZIKV RNA level in the infected cells. A standard curve was included for calculating the absolute RNA level. Total ZIKV RNA is shown as copies in Log10 per well of a 24-well plate. (**C**) MG132 treatment restores the Numb protein levels in ZIKV-infected cells. Vero cells were infected with ZIKV at an MOI of 1 and 30 hpi and treated with MG132 or NH4Cl at a final concentration of 10 µM and 10 mM, respectively, for 6 h. The solvent DMSO was included as a control in the first two lanes. (**D**) ZIKV infection shortens the Numb protein half-life. The addition of cycloheximide to the Vero cells infected with ZIKV at an MOI of 10 for 24 h and treated with MG132 for 6 h. Mock-infected cells were treated similarly and included for control. (**E**) ZIKV infection increases the level of Numb protein polyubiquitination. ZIKV-infected Vero cells were harvested for immunoprecipitation with Numb antibody, followed by Western blotting with antibodies against ubiquitin (Ub) and Numb. Western blotting of the IP input of the cell lysate was also performed as a control. Representative data of three independent experiments are shown.

**Figure 4 viruses-15-01258-f004:**
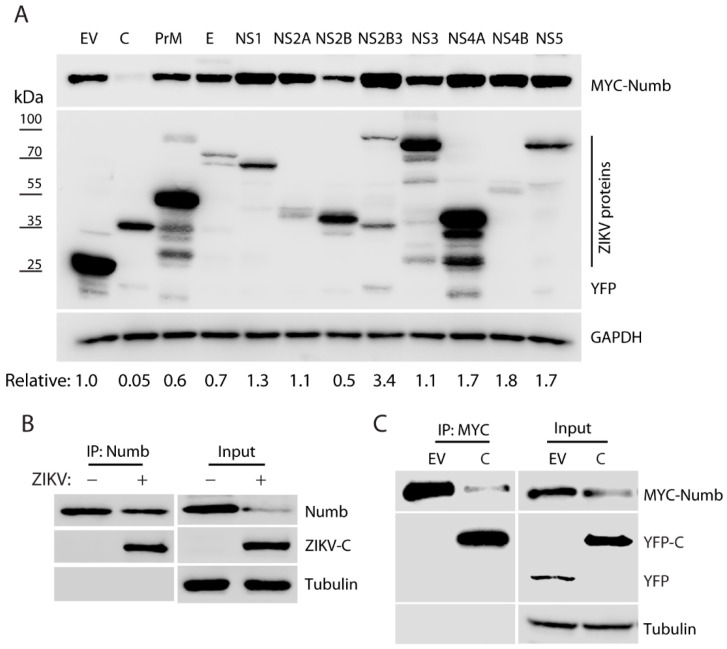
ZIKV capsid protein induces Numb reduction. (**A**) Screening of the ZIKV proteins for inducing Numb reduction. Co-transfection of HEK293 cells was performed with the plasmids encoding the individual ZIKV proteins and MYC-Numb. At 48 h post-transfection, the cells were lysed for WB analysis. Relative Numb levels are indicated below the images. E.V.: empty vector. (**B**) ZIKV capsid is present in the Numb co-IP precipitates. The IP input was included in WB for control. (**C**) The capsid protein is present in the MYC co-IP precipitates. HEK293 cells were co-transfected with MYC-Numb and YFP-C plasmids. E.V.: empty vector. Representative data of three independent experiments are shown.

**Figure 5 viruses-15-01258-f005:**
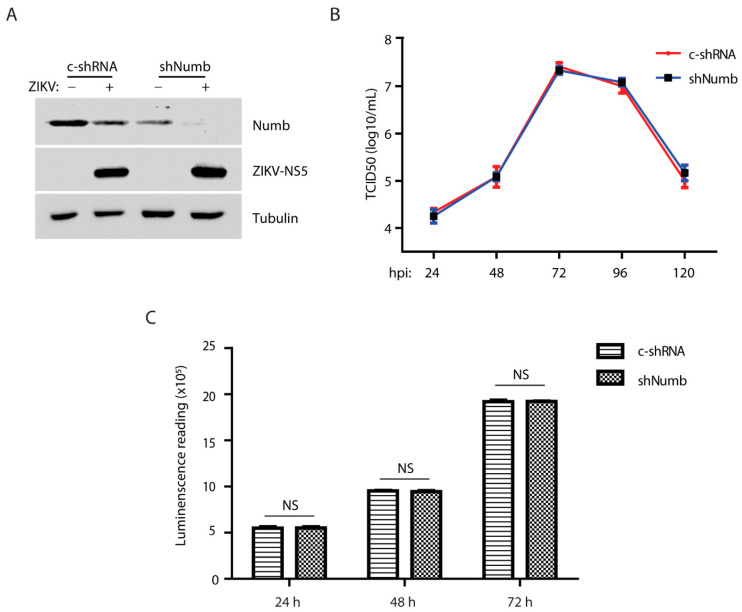
*Numb* knockdown has minimal effect on ZIKV replication. (**A**) *Numb* knockdown in Vero cells has no significant effect on ZIKV replication compared with the control shRNA. The cells were transduced with the recombinant retrovirus containing control shRNA (C-shRNA) or shRNA against *Numb* (shNumb) three times, followed by inoculation with ZIKV at an MOI of 1, and harvested at 48 hpi. (**B**) ZIKV titers from infected Vero cells at different time points. Vero cells were infected by ZIKV at an MOI of 0.01 and harvested at different time points. (**C**) *Numb* knockdown has minimal effect on cell viability. NS: no significant difference.

**Table 1 viruses-15-01258-t001:** List of primers used in this study.

Primer ^a^	Sequences (5′ to 3′) ^b^	Target Gene/Vector
ZIKV-RR-F	AARTACACATACCARAACAAAGTGGT	NS5
ZIKV-RR-R	TCCRCTCCCYCTYTGGTCTTG	NS5
Numb-F1	C*GAATTC*AACAAATTACGGCAAAGTTT	Numb
Numb-R1	G*CTCGAG*TTAAAGTTCAATTTCAAACG	Numb
Numb-RR-F1	GCTACCACCAGTCCCTTCTT	Numb
Numb-RR-F1	GTGCCTGTAGGAACCTCTGT	Numb
shNUMB1-F	GATCCGGAATAAATATTATATATATTCAAGAGATATATATAATATTTATTCCTTTTTTG	shNumb
shNUMB1-R	AATTCAAAAAAGGAATAAATATTATATATATCTCTTGAATATATATAATATTTATTCCG	shNumb
shNUMB2-F	GATCCGCTCTATAGAGAATATATATTCAAGAGATATATATTCTCTATAGAGCTTTTTTG	shNumb
shNUMB2-R	AATTCAAAAAAGCTCTATAGAGAATATATATCTCTTGAATATATATTCTCTATAGAGCG	shNumb
shNUMB3-F	GATCCGAATAAATATTATATATAATTCAAGAGATTATATATAATATTTATTCTTTTTTG	shNumb
shNUMB3-R	AATTCAAAAAAGAATAAATATTATATATAATCTCTTGAATTATATATAATATTTATTCG	shNumb

^a^ F: forward primer, R: reverse primer. ^b^ The italicized letters indicate restriction enzyme cleavage sites for cloning.

## Data Availability

Not applicable.

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
