# Peer review of "Zika Virus Induces Degradation of the Numb Protein Required through Embryonic Neurogenesis"

_viruses, 2023, doi:10.3390/v15061258_

Round 1
Reviewer 1 Report
Comments to authors
The manuscript by He et al. investigated a molecular mechanism by which the Zika virus (ZIKV) causes neuropathogenesis in cultured cells. The authors showed that ZIKV induced the degradation of the Numb protein which is required for neurogenesis. Precisely, the ZIKV capsid protein interacted with the Numb protein and induced the Numb protein degradation through the ubiquitin-proteasome pathway in cultured cells. These data may contribute to a better understanding of neuropathogenesis during ZIKV replication. Overall, the data are clean, and the manuscript is well-written. However, there are some criticisms that are needed to address.
Major comments
1. All data presented are derived from cell culture experiments, hence may not truly depict natural phenomena. Validating findings by conducting in vivo studies can certainly strengthen the study.
2. The authors mainly relied on Vero cells for most of the experiments, which are monkey kidney cells and do not represent the actual tropism for ZIKV. Since ZIKV primarily targets neuronal cells, the use of neuronal cells would have been a better choice for this entire study. I would suggest authors validate some key findings using human neuronal cells.
3. ZIKV infection induces global translational arrest in host cells without affecting transcription (PMID: 28592527, 36546404). Is the degradation of the Numb protein due to global translational arrest? I feel that this is one of the weakest aspects of the study, and the authors need to rule out/clarify it with some additional complementary experiments.
4. The introduction section lacks relevant literature about known mechanisms of ZIKV neuropathogenesis.
Minor comments
1. In Figure 1C, replace Hela with HeLa.
Author Response
- All data presented are derived from cell culture experiments, hence may not truly depict natural phenomena. Validating findings by conducting in vivo studies can certainly strengthen the study.
Answer: Thanks for your advice. Our study was conducted on multiple cell lines and showed the reduction of the Numb protein in ZIKV-infected cells. Validating findings in an animal model can strengthen the study, which is true for all cell culture-based studies. But cell culture studies can provide better-controlled information than animal studies. When possible, we will consider animal studies to confirm our findings in the future.
- The authors mainly relied on Vero cells for most of the experiments, which are monkey kidney cells and do not represent the actual tropism for ZIKV. Since ZIKV primarily targets neuronal cells, the use of neuronal cells would have been a better choice for this entire study. I would suggest authors validate some key findings using human neuronal cells.
Answer: Thank you for your suggestion. We used multiple cell lines, including human neuroblastoma SK-N-SH cells to confirm the reduction of the Numb protein by ZIKV infection in Figure 1, which shows it is not a cell line-specific result. For the convenience of experimental operation, we used Vero and HEK293 cells to determine the molecular mechanism of ZIKV-induced Numb protein reduction. We expect that the mechanism should also extrapolate to human neuronal cells. The student doing this project left over two years ago. We will consider such studies to confirm our findings in the future.
- ZIKV infection induces global translational arrest in host cells without affecting transcription (PMID: 28592527, 36546404). Is the degradation of the Numb protein due to global translational arrest? I feel that this is one of the weakest aspects of the study, and the authors need to rule out/clarify it with some additional complementary experiments.
Answer: Our data does not support the assumption. We are aware of the earlier papers and cited one (PMID: 28592527) in our reference. Our earlier publications showed that ZIKV infection induced elevation of KPNA6 (PMID: 29444946) and reduction of KPNA2 (PMID: 32924767), but has no effect on KPNA1. The viral infection takes advantage of some cellular proteins while inhibiting some others or having no effect on the rest, which cannot be explained by a global translational arrest. We think translational arrest may occur at a certain time after virus infection but depending on the context and viral protein levels because viral proteins are translated using cellular machinery.
- The introduction section lacks relevant literature about known mechanisms of ZIKV neuropathogenesis.
We have modified the introduction section. See line 41-43.
Minor comments
- In Figure 1C, replace Hela with HeLa.
Answer: Corrected.
Reviewer 2 Report
The manuscript “Zika virus induces the degradation of the Numb protein that is required through embryonic neurogenesis” is dedicated to the role of the Numb protein in Zika virus infection, a Flavivirus known to cause neurodevelopmental abnormalities, such as microcephaly. The Numb protein is known to regulate asymmetric cell division which is a crucial process for nervous system embryonic development. However, the current research is based on model cell lines and is focused purely on the mechanistic interactions of the Numb protein with ZIKV proteins.
The manuscript presents comprehensive research and shows clear results. But after careful review, some issues arise. The first issue is that the raw Western blotting membranes do not contain a ladder and do not show the repetitions of the experiments. As I understand, the authors show only representative membranes. But it would be worth following the reproducibility of the analysis. Secondly, the IP experiment does not seem to be repeated, at least, there is no indication of the repetitions in the Figure description/Results. Is it just a single observation? Next, the Figure 1 presents Numb expression in different cell lines and at different time points. Why the specific time points have been chosen? What is the rationale behind it as the time points seem to be specific for specific cell lines. Have there been some trial experiments done that are not referred in this research? Or is there literature data that should be referenced here? It seems that the observation of Numb expression really should be done in some narrow temporal window. The Figure 2 tries to clarify this issue, but the design of the experiment is weird. Instead of MOI=10, you infect cells with MOI = 1 and see some Numb reduction at 20 and 40 hpi, but then suddenly, you select 24 phi to test different MOI. Is it that these experiments have been done actually before the ones, presented in Fig. 1? The same question is about NS4B and NS5 used to show the presence of ZIKV on different figures. What is the rationale behind the choices? Why not uniform throughout the research?
Later, describing the protein stability experiments authors mention some densitometry (Line 214). But the Figure 3 only shows raw WB membranes. The common way to show this type of experiments is actually some kind of curve or graph. The authors somehow have determined the ratio of ubiquitination, but do not show any quantification. And again, the question about repetitions arises.
Also, to check Numb RNA level authors address 30 phi of infection which has not been covered in previous experiments, no explanation to that either. I would also strongly recommend adding the description of the copy number determination to Materials and Methods, as just copy number per well is not enough.
And finally, in the experiment with the knock-down of the Numb gene, what was the MOI and the ratio with the cells? It seems the authors were starting with low density and small MOI, if the cells have survived for 5 days. Would these parameters influence the interaction of viral proteins with the Numb? And the knockdown was checked 2 days after transduction but the experiment lasted 5 days. Is the knockdown stable?
So, to sum up the findings of the presented research, I would expect the authors to design their experiments more consistently and transfer their findings to cells of neuronal origin (such as SK-N-SH). The nature of the Numb regulation by the ZIKV is not clear after all, because the function of the protein under study does not seem to influence the viral infection.
The overall impression after reading the article is good, and I would be happy to recommend this research for publication in such a reputable journal as Viruses after some minor corrections and explanations.
The article is written in clear English and almost does not contain typos or unclear sentences.
The minor corrections are as follows:
Line 3 – thorough?
Line 76 – Numb plasmid?
Table 1 – the Numb-R1 primer is misaligned.
Line 99 – what do "alphabets" mean? You mean letters?
Line 111-113 – the description of the ChemiDoc is inconsistent. Is it Bio-Rad or Hercules? It is Bio-Rad usually.
Line 309-310 – that there might have post-translational? The sentence needs correction.
Author Response
Thank you for your thoughtful comments and feedback. Here are our responses to each question:
- Why raw WB membrane do not contain a ladder and do not show the repetitions of the experiments.
Answer: We included a prestained protein ladder in SDS-PAGE, which is convenient for observation during gel electrophoresis and transfer. The WB membranes were scanned for luminescence signals after blotting with a Bio-Rad Chemi-Doc Imaging System. The color protein ladder was scanned right afterward using Epi white light. We then overlay the images to assess the molecular weight of the bands visible. For submission, only the raw WB images are submitted, instead of the ladder images because they were not taken from the same channel. Also, only representative images of multiple experiments are submitted.
- Secondly, the IP experiment does not seem to be repeated, at least, there is no indication of the repetitions in the Figure description/Results.
Answer: To confirm our findings, we conducted the IP experiment at least three times. We’re sorry that we forgot to mention the repetition in the manuscript, as it is a common practice in biology study. We added this information in the figure legend of Figures 3 and 4: “Representative data of three independent experiments are shown.”
- Next, the Figure1 presents Numb expression in different cell lines and at different timepoints. Why the specific time points have been chosen? What is the rationale behind it as the time points seem to be specific for specific cell lines. Have there been some trial experiments done that are not referred in this research? Or is there literature data that should be referenced here?
Answer: In our earlier study of ZIKV effect on KPNA2 (PMID: 32924767), we used variable MOI and different time points in different cell lines to confirm the findings. Also, ZIKV replication seems faster in Vero than in HeLa cells and is slow in SK-N-SH cells, which may also be because the cell growth behaves differently among the cell lines. So, we determined the harvesting time according to virus replication in the cell lines. We added a citation of the paper (PMID: 32924767) at the beginning of the Results section in the revision. In the beginning, we used high MOI (10) to ensure that most cells were infected with the ZIKV virus. With a high level of viral infection, we confirmed the significant reduction of the Numb protein by ZIKV infection. However, a high MOI can lead to cell death if cell incubation is extended long. Considering this potential effect of high MOI, we studied the time-dependent effect of ZIKV on the Numb protein with a lower MOI (1). Cell harvesting around 24 hpi is optimal when different MOIs were tested because of minimal cell death even with the high MOI.
For the detection of ZIKV proteins, we ordered antibodies against several ZIKV proteins. NS5 antibody was used most often. Other antibodies, such as those against NS4B and E, were also used occasionally. We agree with the reviewer that it would be best to be consistent in all figures. However, the detection of different target proteins enhances our confidence in the presence of virus infection.
- Later, describing the protein stability experiments authors mention some densitometry (Line 214). But the Figure 3 only shows raw WB membranes. The common way to show this type of experiments is actually some kind of curve or graph. The authors somehow have determined the ratio of ubiquitination, but do not show any quantification. And again, the question about repetitions arises.
Answer: Thank you for the comments. Figure 3 shows the WB images of a representative experiment. In future studies, we will consider using the presentation of the curve or graph. We added the ratio of ubiquitin blotting in Figure 3E. “Representative data of three independent experiments are shown.” is added to the figure legend.
- Also, to check Numb RNA level authors address 30 phi of infection which has not been covered in previous experiments, no explanation to that either. I would also strongly recommend adding the description of the copy number determination to Materials and Methods, as just copy number per well is not enough.
Answer: Since the ZIKV-induced reduction of the Numb protein increases along with the extension of infection shown in Figure 2, we harvested the cells at a time point between those tested (20 and 40 hpi) for the RNA level determination. The reason was that we thought ZIKV proliferation at 30 hpi is better than 20 hpi and the overall cell condition at 30 hpi is better than the 40 hpi. So, striking a balance the 30 hpi was used in this experiment. However, other time points can also be used and are expected to yield similar results, showing a minimal effect on the Numb RNA.
We added the copy number description to the Materials and Methods. See line 99-102.
- And finally, in the experiment with the knock-down of the Numb gene, what was the MOI and the ratio with the cells? It seems the authors were starting with low density and small MOI, if the cells have survived for 5 days. Would these parameters influence the interaction of viral proteins with the Numb? And the knockdown was checked 2 days after transduction but the experiment lasted 5 days. Is the knockdown stable?
Answer: For Figure 5A, the cells were infected at an MOI of 1. For 5B, the cells were infected at an MOI of 0.01. We updated the figure legend. Since we did not see any change in ZIKV NS5 protein level after the Numb knockdown shown in Figure 5A, we conducted virus titration for samples of multiple days to confirm the finding. We don’t think the parameters would influence the interaction of Numb and viral proteins. Increasing MOIs has an incremental effect on Numb reduction shown in Figure 2. The knockdown was stable because GFP expression from the retrovirus vector for shRNA was observed for multiple days and even several passages after transduction.
- So, to sum up the findings of the presented research, I would expect the authors to design their experiments more consistently and transfer their findings to cells of neuronal origin (such as SK-N-SH). The nature of the Numb regulation by the ZIKV is not clear after all, because the function of the protein under study does not seem to influence the viral infection.
Answer: Thanks for your suggestion. We agree that it is better to have a consistent design and test all findings in cells of neuronal origin. Unfortunately, the COVID pandemic had some impact on the project, and the graduate student had to move on. Our current data show that Numb knockdown has no effect on ZIKV replication. The Numb reduction may contribute to ZIKV pathogenesis. If possible, in the future, we’ll do research on Numb protein in vivo to determine its role in ZIKV pathogenesis.
- The overall impression after reading the article is good, and I would be happy to recommend this research for publication in such a reputable journal as Viruses after some minor corrections and explanations.
Answer: Thank you for your comments. We have revised the manuscript accordingly to address the concerns.
Comments on the Quality of English Language
The article is written in clear English and almost does not contain typos or unclear sentences.
The minor corrections are as follows:
Line 3 – thorough? It is “through”
Line 76 – Numb plasmid? Corrected, thanks.
Table 1 – the Numb-R1 primer is misaligned. Corrected, thanks.
Line 99 – what do "alphabets" mean? You mean letters? Corrected, thanks.
Line 111-113 – the description of the ChemiDoc is inconsistent. Is it Bio-Rad or Hercules? It is Bio-Rad usually. Answer: It is the Bio-Rad laboratories in Hercules, CA. To make it clear, we rewrote the sentence: the signal was recorded digitally using a Bio-Rad ChemiDoc X.R.S. imaging system with the QuantityOne Program, version 4.6 (Bio-Rad Laboratories, CA).
Line 309-310 – that there might have post-translational? The sentence needs correction.
Corrected. The sentence reads “…which suggests that post-translational cleavage might have happened to them.” See lines 318-320.
Reviewer 3 Report
Zika virus induces the degradation of the Numb protein that is required through embryonic neurogenesis
In this publication, the authors detail how ZIKV infection causes a decrease in the Numb related to embryonic neurogenesis.
They examine the Numb status in three cell lines (Vero, Hela, and SK-N-SH). They also compare two ZIKV lineages (Asian and African). They discover that the Asian lineage contributes more to the reduction of Numb in Vero cells. Following that, they find that the Numb decrease is related to the Moi of ZIKV, with the higher the Moi, the more significant the Numb reduction.
They then investigate which pathway ZIKV uses to diminish Numb cellular levels via the MG132 proteasome inhibitor, which prevents ubiquitin-conjugated protein degradation. They demonstrate that ZIKV decreases Numb via the ubiquitin-proteasome pathway. The author's next show that the capsid protein is responsible for reducing Numb by transfecting in HEK293 cells; they also immunoprecipitate it and detect the union with Numb, which supports their findings.
They carried out a Numb KO system on Vero cells after infecting them with ZIKV and discovering that a Numb KO does not affect ZIKV replication.
Major corrections
1. Numb's background is missing from the intro. Are there studies on other viral infections? Why was it decided to study Numb and not another protein related to embryonal neurogenesis?
2. Figure 1: Are there differences in Numb expression between different cells? Why were both ZIKV lineages only compared in Vero cells and not in the other cell lines?
3. Perform a statistical analysis and bar graph for the optical densities of the Western Blots in all figures.
4. Discuss why the Asian lineage reduces Numb more (compared to the African) and how this result compares with human disease.
5. Figure 2: Why analyze Numb reduction only in Vero cells, not neuroblastoma cells?
6. In Figure 4, again, statistics analysis is needed to compare better the different ZIKV proteins and the reduction of Numb. It seems that NS2B also reduces Numb. Perhaps statistical analysis can help.
7. Although Numb KO cells have no effect on replication with ZIKV. What would happen with a Numb overexpression system, through transfection of a plasmid with the Numb sequence and the infection with ZIKV to different Moi?
Minor corrections
The first paragraph of the introduction should be revised to make its reading more fluent.
Your work is very interesting! I suggest checking the grammar of the text
Author Response
Thank you for your comments. Here are answers for the major corrections:
- Numb's background is missing from the intro. Are there studies on other viral infections? Why was it decided to study Numb and not another protein related to embryonal neurogenesis?
Answer: The Numb protein is unique in its function in embryonal neurogenesis. It was selected because of its unique role and ZIKV’s neural pathogenesis. We modified the introduction to help to understand the background (see lines 41-47). The Numb in other virus infections was described in the Discussion. “The Numb protein is needed for hepatitis C virus entry as RNAi-mediated knockdown of the Numb inhibits the virus entry. However, Numb inhibition activates Notch signaling, which assists the transcription of covalently closed circular DNA (cccDNA) of the hepatitis B virus.” See lines 343-346.
- Figure 1: Are there differences in Numb expression between different cells? Why were both ZIKV lineages only compared in Vero cells and not in the other cell lines?
Answer: We don’t know if there are differences in Numb expression between different cells. But Different cell lines have different susceptivity to ZIKV infection. We use Vero cells most often in the ZIKV study and, thus, used this cell line to test the two ZIKV lineages on Numb reduction. We expect they have a similar effect on Numb in other cell lines. ZIKV PRABC59 strain belongs to the Asian lineage, and the MR766 strain belongs to the African lineage. Since we already confirmed the Numb reduction happened with both strains’ infection in Vero cells, Numb reduction by the MR766 strain is expected to happen in other cell lines.
- Perform a statistical analysis and bar graph for the optical densities of the Western Blots in all figures.
Answer: Thank you for the suggestion. Western blotting is a semi-quantitative method to assess protein expression. Showing representative images of multiple experiments is a well-accepted practice in publications. As explained above, the graduate student doing the project left, and we are unable to accomplish the suggested analysis and bar graph. But in the future, we will pay attention to this suggestion in our studies.
- Discuss why the Asian lineage reduces Numb more (compared to the African) and how this result compares with human disease.
Answer: We added a discussion about the possible reasons. See lines 290-295.
- Figure 2: Why analyze Numb reduction only in Vero cells, not neuroblastoma cells?
Answer: Since ZIKV reduces the Numb protein in all the cell lines tested, we just used Vero cells that we frequently use in the lab for the study to show the time and dose-dependent effect. It is expected that the virus has a similar effect in neuroblastoma cells.
- In Figure 4, again, statistics analysis is needed to compare better the different ZIKV proteins and the reduction of Numb. It seems that NS2B also reduces Numb. Perhaps statistical analysis can help.
Answer: Thank you for your suggestion. As explained above, the representative images are shown for the screening and serve the purpose of identifying the ZIKV protein for Numb reduction. Statistical analysis will enhance the finding but won’t change the conclusion. Yes, NS2B also induced a reduction of the Numb protein, but much less than the C. So, we focused on C to confirm the effect and conducted IP to show the interaction with Numb.
- Although Numb KO cells have no effect on replication with ZIKV. What would happen with a Numb overexpression system, through transfection of a plasmid with the Numb sequence and the infection with ZIKV to different Moi?
Answer: Our data show that Numb knockdown has no effect on ZIKV replication, which suggests that the Numb is not directly involved in the ZIKV life cycle. We have no idea if overexpression of Numb has any effect on the virus replication. But it is unlikely that the presence of more Numb protein would affect the virus replication. We don’t think a different MOI would yield a different result in this case because the Numb protein does not involve in the virus life cycle.
Minor corrections
The first paragraph of the introduction should be revised to make its reading more fluent.
We have revised it. Thanks.
Round 2
Reviewer 1 Report
The manuscript is now suitable for publication.
Author Response
Thank you for your review!
Reviewer 3 Report
Line 29 omit “(“capsid
Line 58-59 “Drosophila sp.”
Line 226-227, adjust font size
Homogenize the use of “Numb” or “Numb” throughout the writing
Check that the bibliography is the one required by Viruses
None
Author Response
Thank you for the comments. All have been taken care of.